# Temperature cues are integrated in a flexible circadian neuropeptidergic feedback circuit to remodel sleep-wake patterns in flies

Xin Yuan[1,2,3], Hailiang Li[2,3,4], Fang Guo[1,2,3,4]*

1 Department of Neurology of Children's Hospital and School of Brain Science and Brain Medicine, Zhejiang University School of Medicine, Hangzhou, China, 2 MOE Frontier Science Center for Brain Research and Brain-Machine Integration, State Key Laboratory of Brain-machine Intelligence, Zhejiang University, Hangzhou, China, 3 NHC and CAMS Key Laboratory of Medical Neurobiology, Zhejiang University, Hangzhou, China, 4 Department of Neurobiology, Department of Neurology of Sir Run Run Shaw Hospital and School of Brain Science and Brain Medicine, Zhejiang University School of Medicine, Hangzhou, China

* gfang@zju.edu.cn

## Abstract

Organisms detect temperature signals through peripheral neurons, which relay them to central circadian networks to drive adaptive behaviors. Despite recent advances in *Drosophila* research, how circadian circuits integrate temperature cues with circadian signals to regulate sleep/wake patterns remains unclear. In this study, we used the FlyWire brain electron microscopy connectome to map neuronal connections, identifying lateral posterior neurons LPNs as key nodes for integrating temperature information into the circadian network. LPNs receive input from both circadian and temperature-sensing neurons, promoting sleep behavior. Through connectome analysis, genetic manipulation, and behavioral assays, we demonstrated that LPNs, downstream of thermo-sensitive anterior cells (ACs), suppress activity-promoting lateral dorsal neurons LNds via the AstC pathway, inducing sleep Disrupting LPN-LNd communication through either *AstCR1* RNAi in LNds or in an *AstCR1* mutant significantly impairs the heat-induced reduction in the evening activity peak. Conversely, optogenetic calcium imaging and behavioral assays revealed that cold-activated LNds subsequently stimulate LPNs through NPF-NPFR signaling, establishing a negative feedback loop. This feedback mechanism limits LNd activation to appropriate levels, thereby fine-tuning the evening peak increase at lower temperatures. In conclusion, our study constructed a comprehensive connectome centered on LPNs and identified a novel peptidergic circadian feedback circuit that coordinates temperature and circadian signals, offering new insights into the regulation of sleep patterns in *Drosophila*.

## Introduction

Maintaining appropriate sleep and activity levels across different seasonal environments is crucial for animal health and survival [1–5]. However, the precise neural mechanism underlying the adaptive regulation of sleep/wake behavior in response to external cues, like hibernation

**Funding:** This work is supported by funding from the National Key Research and Development Program of China (2019YFA0802400), the National Natural Science Foundation of China (31970941, 32171008, 32471210), the Zhejiang Provincial Outstanding Youth Science Foundation (LR20C090001), the Non-profit Central Research Institute Fund of Chinese Academy of Medical Sciences (2023-PT310-01), the Fundamental Research Funds for the Central Universities (2023ZFJH01-01, 2024ZFJH01-01) and the Fundamental Research Funds for the Central Universities to F.G. The funders had no role in study design, data collection and analysis, decision to publish, or preparation of the manuscript.

**Competing interests:** The authors have declared that no competing interests exist.

**Abbreviations:** AC, anterior cell; ATR, all-trans-retinal; ChAT, choline acetyltransferase; DN, dorsal neuron; EM, electron microscopy; LN, lateral neuron; nAChR, nicotinic acetylcholine receptor; SD, standard deviation; SEM, standard error of the mean.

behavior in mammals, remains incompletely understood [6–9]. In fruit flies, environmental temperature changes induce distinctly adaptive modifications in sleep/wake profiles [10]. The circadian neuron system receives temperature signal and integrates this information with circadian information to orchestrate suitable behavioral responses [11]. In *Drosophila*, the fundamental circadian neurons are categorized into groups of lateral neurons (LNs) and dorsal neurons (DNs), based on their anatomy [12]. Among these, DN1, LPN, LNd, and DN3, are reported to be temperature-sensitive [13–17].

Peripheral temperature-sensitive cells, including cold cells, hot cells, and anterior cells (AC), respond to thermal changes and relay signals to higher brain regions [18,19]. The secondary thermosensory projection neurons process the information about prolonged cold temperatures and relay signals to higher brain centers, including the circadian neuron DN1a, which reshapes normal daytime sleep under cold conditions [14]. Conversely, high temperatures activate ACs via Transient Receptor Potential A1 (TrpA1), sending signals to DN1ps [15] and LPNs [16]. DN1p promotes nighty wakefulness by inhibiting DH44$^+$ neurons through a CNMa-dependent mechanism [15]. Those cues suggest that the perception of temperature information by circadian neurons is critical for the plastic regulation of arousal and sleep for survival. However, the dynamic communication among circadian neurons and their adjustments in response to temperature changes remains incompletely understood.

The recent studies have highlighted a strong connection between LPNs and heat-sensitive ACs [16]. Additionally, LPNs have been reported to promote daytime sleep [20], while other research suggests a slight inhibitory effect on sleep behavior [21]. These conflicting results prompted us to investigate the potential communication within the circadian neuron network, focusing on the role of LPNs in temperature-sensitive sleep regulation.

Advances in brain structure reconstruction using electron microscopy (EM) have made it possible to clearly map the neural networks of model organisms like *Drosophila* [22–24]. Detailed dissection of neural connectomes based on EM data has significantly advanced our understanding of neural circuits involved in sensory processing, such as olfactory and visual systems [25,26], as well as innate behaviors [27,28]. Recently, the synaptic connectome of *Drosophila* circadian neuron revealed an extensive synaptic connectivity within the network, uncovering novel light input pathways and key neurosecretory output cells [29].

In our study, we utilized *Drosophila* FlyWire [23] connectomics analysis to demonstrate that LPNs primarily receive input from both circadian and temperature-sensing neurons. Additionally, we identified a circadian feedback circuit that dynamically modulates the sleep-wake profile in response to varying temperature conditions. At elevated temperatures, LPNs are highly active, receiving heat signals from TrpA1$^+$ ACs and promoting sleep by inhibiting wake-promoting LNds via the AstC signaling pathway. Conversely, LNds receive cold signals from DN1a and become active at lower temperatures [17], then activate LPNs through the NPF pathway, establishing a feedback inhibition mechanism. This integration of temperature information into a circadian feedback circuit enables adaptive behavioral responses to different thermal environments. Our findings provide significant insights into the dynamic interplay between temperature cues and circadian signaling within the central circadian circuit.

## Results

### LPN promote evening sleep via AstC under high temperature condition

We initiated our investigation by exploring the role of LPN in sleep regulation. Utilizing the LPN-specific split Gal4 line (*LPN-spGal4*) [30,31], we identified three LPN-like neurons in each hemibrain, as well as several neurons in the optic lobe (Fig 1A). To verify the identity of these 3 neurons as circadian LPNs, we performed co-staining with an antibody against the

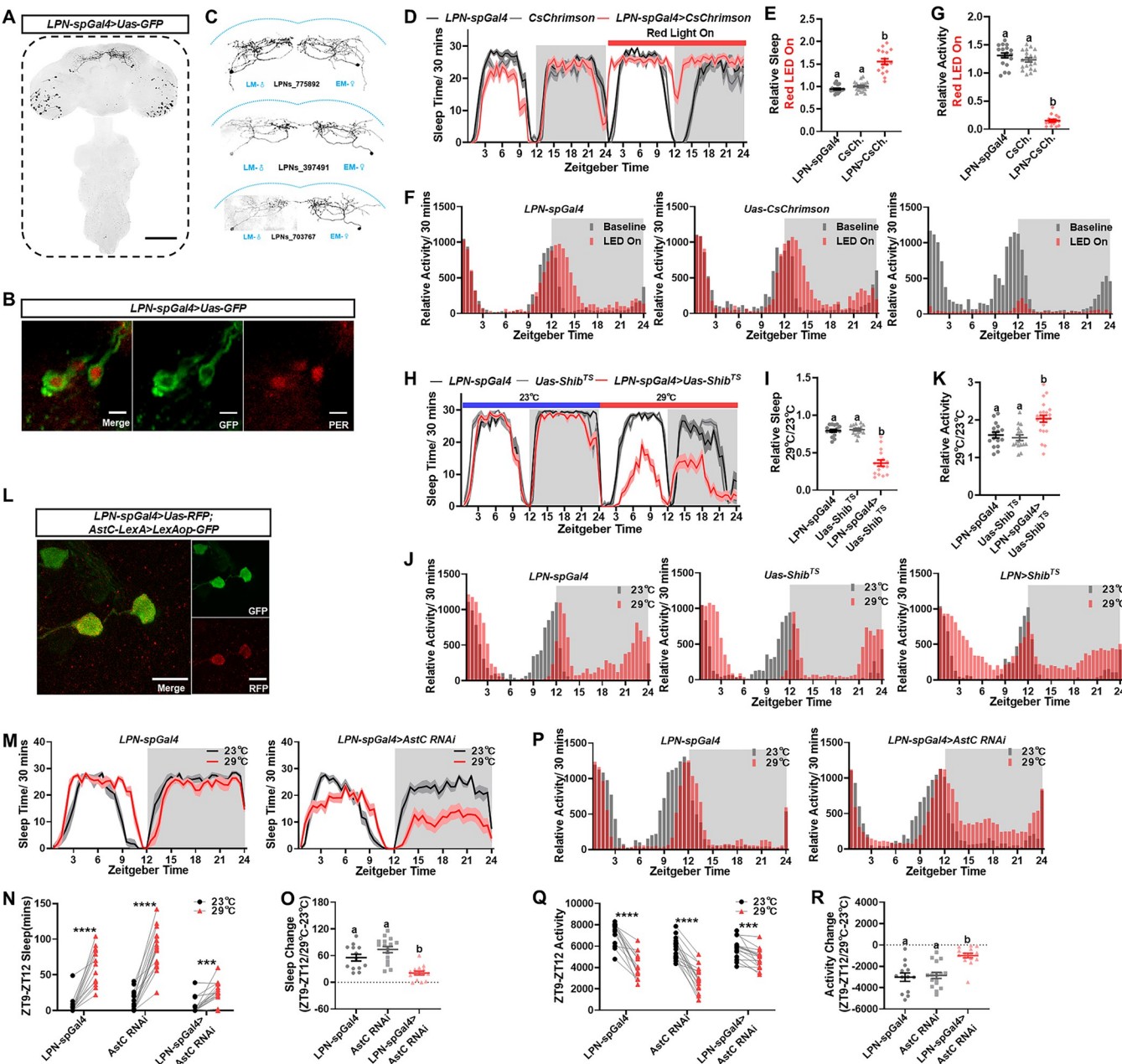

**Fig 1. The circadian LPN promote sleep through AstC pathway.** (A) Confocal images showing LPN-spGal4 expression in the brain and ventral nerve cord (*LPN-spGal4>Uas-GFP*; green). Scale bars, 100 μm. (B) Co-staining of PER (red) with LPNs (*LPN-spGal4>Uas-GFP*) confirms their circadian identity. Scale bars, 10 μm. (C) Left: Single-cell morphology of LPNs via LM. Right: Corresponding single LPN morphology from Flywire EM reconstruction (v783) [23]. (D, E) Optogenetic activation of LPNs using CsChrimson (*LPN-spGal4>CsChrimson*; red line) results in a significant increase in sleep compared to baseline. (F, G) LPN activation by CsChrimson also causes a significant reduction in daily locomotion compared to baseline. (H, I) Thermogenetic inhibition of LPNs with Shibire[TS] (*LPN-spGal4>Shib[TS]*; red line) leads to a significant decrease in sleep relative to baseline. (J, K) Inhibition of LPNs with Shibire[TS] results in a significant increase in daily locomotion compared to baseline. (L) AstC neurons (*AstC-LexA>LexAop-GFP*; green) are colocalized with LPNs (*LPN-spGal4>Uas-RFP*; red). Scale bar, 10 μm. (M–O) *AstC* knockdown in LPNs (*LPN-spGal4>Uas-AstC RNAi*) blocks the high-temperature-induced evening sleep increase (ZT9-ZT12) (red line, 29°C) relative to baseline (black line, 23°C). (P–R) *AstC* knockdown in LPNs also prevents the high-temperature-induced decrease in evening locomotion (ZT9-ZT12) (red bar, 29°C) relative to baseline (black bar, 23°C). Data (E, G, I, K, O, R) were analyzed using Welch's one-way ANOVA with multiple comparisons, and letters a, b indicate statistical significance, $P < 0.05$. Data (N, Q) were quantified with paired $t$ test with Welch's correction. ***$P < 0.001$, ****$P < 0.0001$. The raw data in this figure including D, E, F, G, H, I, J, K, M, N, O, P, Q, and R can be found in S1 Data. EM, electron microscopy; LM, light microscopy.

circadian protein PER, confirming that the *LPN-spGal4* line labels circadian LPNs (Fig 1B). For morphological analysis, we sparsely labeled each LPN individually using MultiColor FlpOut technology in male flies [32]. Comparison with female EM data from the FlyWire brain connectome [23,33] revealed minimal gender-based morphological differences in circadian LPNs (Fig 1C), reinforcing the reliability of using FlyWire to investigate connectivity.

To investigate the function of LPNs, we conducted optogenetic and thermogenetic experiments to manipulate LPN activity and assessed sleep behavior using the Flybox [34]. Activation of LPNs with the red light-gated cation channel *CsChrimson* [35] significantly increased sleep during the red LED day (Fig 1D and 1E), leading to reduced morning and evening locomotor peaks (Fig 1F and 1G). To mitigate potential effects of light on sleep profiles, we also activated LPNs using the thermal channel *dTrpA1* [18,36] and observed a similar increased sleep (S1A and S1B Fig) and reduced locomotion in *LPN>dTrpA1* flies (S1C and S1D Fig). Conversely, inhibition of LPNs using the thermally induced blocking tool *Shibire*[TS] [37] reduced sleep during the restrictive temperature day (Fig 1H and 1I) and broadened the morning and evening peaks (Fig 1J and 1K). Overall, our results robustly support the essential role of LPNs in sleep regulation [20,21].

LPNs have been shown to express the neurotransmitter glutamate [20] and the neuropeptides Allatostatin A (AstA) and Allatostatin C (AstC) [20,21]. We validated these expressions using co-staining and found that LPNs are AstA[+], AstC[+], and cholinergic (Figs 1L, S1E, and S1F). Additionally, we employed an intersectional strategy using *LPN-spGal4* with *ChAT-LexA* or *VGLUT-LexA* to confirm the neurotransmitters. We found that only the intersection of *LPN-spGal4* with *ChAT-LexA* specifically labeled 3 pairs of LPNs (S1G and S1H Fig). Activation of these cholinergic LPNs resulted in a comparable increase in sleep to that observed with *LPN-spGal4* activation (S1I and S1J Fig), indicating that cholinergic LPNs, rather than optic lobe neurons labeled by *LPN-spGal4*, are responsible for promoting sleep. Our results implied that LPNs may utilize neuropeptides or neurotransmitters to modulate sleep behavior.

Recent research has discovered that LPNs receive excitatory input from high-temperature-sensitive dTrpA1[+] ACs, which increases siesta sleep under hot conditions [16]. Our loss-of-function experiments with *Shibire*[TS] showed that LPN-inhibited flies still exhibited a robust evening peak at high temperatures compared to controls (Fig 1J). This indicates that neuropeptides or neurotransmitters released from LPNs during hot conditions can promote evening sleep and inhibit locomotion. Using RNAi against these neuropeptides or neurotransmitters, we found that knockdown of *AstC* in LPNs can significantly block the hot-induced evening sleep increase (Fig 1M–1O) and the evening locomotion decrease (Fig 1P–1R) compared to control groups. These results suggest that the neuropeptide AstC from LPNs is involved in hot-induced sleep and locomotion adjustment.

## Connectomics analysis of LPN neurons reveal major circadian and environmental inputs

Following the systematic annotation of *Drosophila* circadian neurons, targeted connectome analysis has become feasible [29]. As an initial step, we examined the presynaptic connectivity of LPNs. Using the *Drosophila* FlyWire brain connectome [23], we identified 6 LPN neurons by their unique IDs [29]. For each LPN, synaptic connectivity data were categorized into presynaptic and postsynaptic neurons. We aggregated the IDs of all presynaptic neurons and quantified the synaptic projections to each LPN, identifying 134 neurons with ≥5 presynaptic connections to LPNs. We then enumerated the annotated cell types of the top 50 presynaptic neurons (Fig 2A) and summarized the proportional inputs from neurons with ≥5 synapses (Fig 2B).

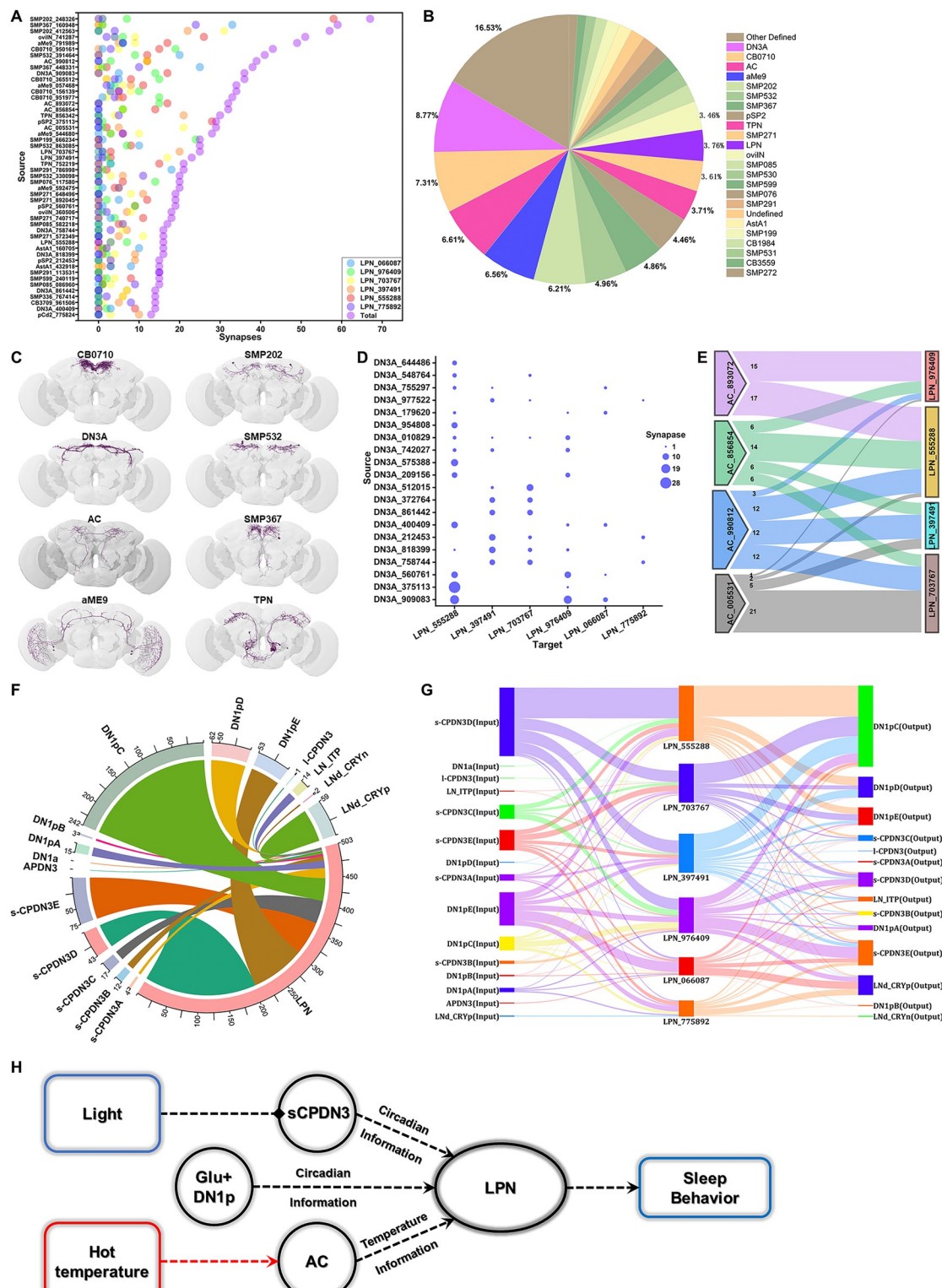

**Fig 2. The connectomics analysis of LPN with Flywire EM database reveal the circadian and environmental input.** (A) Top 50 presynaptic neurons of LPNs identified from FlyWire data set (v783) (https://flywire.ai/) [23]. (B) Percentage input of LPNs from various neuron types with ≥5 synapses, based on Flywire data set (https://flywire.ai/) [23]. (C) EM reconstruction of presynaptic neurons for LPNs shown in (B). (D) Synapse quantification from annotated DN3A to LPN based on public FlyWire data set (v783) (https://flywire.ai/) [23]. (E) Four annotated AC neurons form synaptic connections with 4 LPN neurons. (F) Broad synaptic

interconnectivity between recently annotated circadian cell types and LPNs. Numbers indicate predicted presynaptic inputs from corresponding neuron types. (G) Broad circadian presynaptic inputs and circadian postsynaptic outputs for different single LPNs. The color of the connecting line was kept consistent with the color of the presynaptic neuron. (H) Intrinsic circadian neurons within the brain detect circadian information, while peripheral neurons relay environmental information like temperature changes to the central brain. LPNs integrate circadian signals from DN1p and sCPDN3, and temperature changes from AC neurons to modulate sleep behavior. The raw data in this figure including A, B, D, E, F, and G can be found in S1 Data. AC, anterior cell; EM, electron microscopy.

Although annotation of the public v783 database is ongoing, our analysis revealed that LPNs primarily receive synaptic inputs from DN3A neurons (8.77%), which are associated with circadian information, temperature-sensitive AC/TPN neurons (6.61%/3.71%), and visually sensitive aME9 neurons (6.56%) (Fig 2B). Collectively, these synaptic connections provide circadian and external sensory cues, enabling LPNs to adjust their daily activity in response to changes in the environment. Additionally, the 6 LPNs exhibit strong mutual connectivity, suggesting the formation of a tightly synergistic network (Fig 2B). We verified the identities of each neuron ID using FlyWire's 3D reconstruction of neuronal skeletons (Fig 2C). Further analysis revealed selective DN3A projections to LPNs, with LPN_CB1215 (555288, 397491, 703767, 976409) receiving more projections, while LPN_CB2857 (066087, 775892) received fewer (Figs 2D and S2A). This supports the classification of LPNs into 2 distinct cell types [21,29,38]. In agreement with data from neuprint [16,22], LPN_CB1215 received 6.61% of its synaptic input from 4 heat-sensitive AC neurons (Fig 2E). These findings suggest that LPNs primarily integrate circadian and temperature information to modulate sleep behavior.

Building on a recent study that identified nearly all circadian neurons in FlyWire [29], we conducted a detailed analysis of LPN connectivity within the circadian neuron network, focusing on all annotated IDs for circadian neurons (S2B Fig). Our results revealed that LPNs form primary synaptic connections with DN1p, sCPDN3, and LNd neurons (Fig 2F). While individual LPNs showed slight differences, they generally receive inputs from DN1p and sCPDN3 neurons and project primarily to DN1p, sCPDN3, and LNd neurons (Figs 2G, S2C, and S2D).

Glutamatergic DN1p neurons can inhibit key pacemaker neurons, thereby promoting sleep behavior [34]. Our findings show that RNAi-mediated knockdown of glutamate transmission in DN1ps impairs basal sleep (S3A and S3B Fig) and the glutamate receptor NMDAR1 is expressed in LPNs (S3C Fig). To explore the functional connection between DN1ps and LPNs, we optogenetically activated DN1ps and monitored calcium activity in LPNs. Activation of DN1ps enhances calcium activity in LPNs via Glutamate-NMDAR signaling (S3D and S3E Fig), likely contributing to the sleep-promoting effects of DN1ps. Additionally, the number of DN3 neurons, particularly sCPDN3, identified in FlyWire is notably higher than previously recognized [29], with only APDN3 previously implicated in sleep regulation [39]. Using NeuronBridge [40], we obtained a split-Gal4 line specifically labeling a large population of sCPDN3 neurons (S3F Fig). Our data show that sCPDN3 neurons strongly promote sleep (S3G and S3H Fig) and inhibit activity (S3I and S3J Fig), suggesting an interlinked sleep-promoting DN3-LPN circuit in the fly brain.

In conclusion, our findings suggest that LPNs primarily receive circadian input from DN1p and sCPDN3 neurons, as well as temperature input from AC neurons (Fig 2H). LPNs likely feedback to these circadian neurons, as well as LNd neurons, to modulate sleep behavior.

Ppk$^+$ AC-LPN circuit modulate high temperature induced evening activity change

Our Flywire connection analysis, along with previous paper [16], suggests that high temperature activates ACs, which in turn activate LPNs to modulate the evening locomotion peak. Flywire data confirmed that ACs mainly output to 5 neuron types (S4A and S4B Fig). Among those neurons, 13.66% postsynaptic connections from ACs signal to LPNs (Fig 3A). We

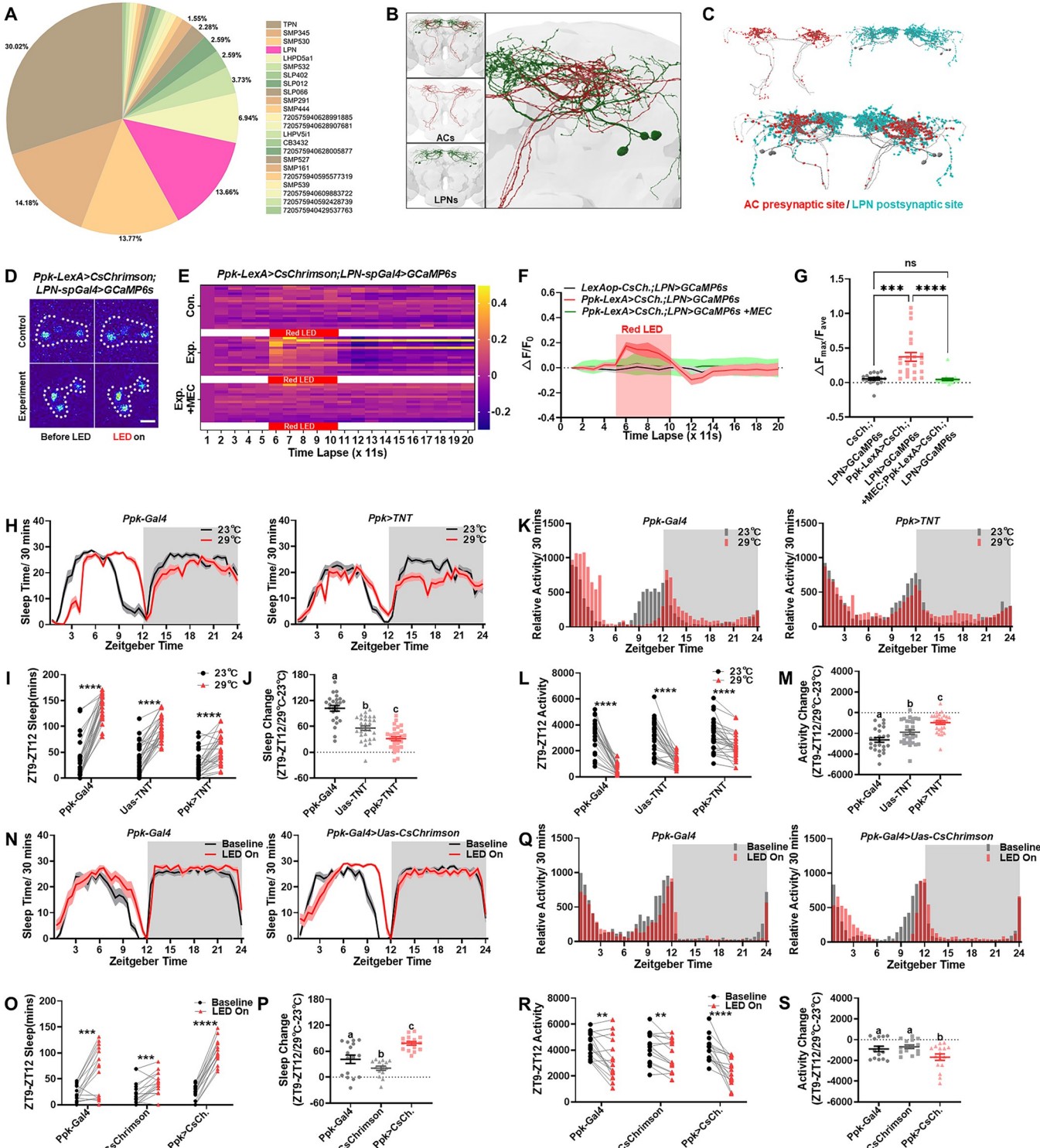

**Fig 3. AC-LPN circuit control high temperature induced increase of evening sleep.** (A) Percentage output of ACs to various neuron types with ≥5 synapses, based on Flywire data set [23]. (B) EM reconstruction of ACs and LPNs, with magnified potential dense synaptic sites (right). (C) Predicted presynaptic sites of ACs and postsynaptic sites of LPNs in Flywire data set [23]. (D–G) Optogenetic activation of ppkACs with CsChrimson (*Ppk-LexA>LexAop-CsChrimson*) increases LPN activity in vivo (D). Scale bars, 10 mm. Ach antagonist-MEC blocks this activation. Heat maps (E), summarized line graphs (F), and normalized maximum calcium fluorescence (G) are shown. Z-stacks of 10 frames with 1.1s intervals. (H–J) The ppk neuron inhibition (*Ppk-Gal4>Uas-TNT*) blocks the high-temperature-induced evening sleep increase (ZT9-ZT12) (red line, 29°C) relative to baseline (black line, 23°C). Quantification of evening sleep (I) and sleep change (J) shown. (K–M) The ppk neuron inhibition also blocks the high-temperature-induced decrease in evening locomotion (ZT9-ZT12) (red bar,

29°C) relative to baseline (black bar, 23°C). Quantification of relative locomotion (L) and locomotion change (M) shown. (N–P) Optogenetic activation of ppkAC using CsChrimson (*Ppk-Gal4>Uas-CsChrimson*) induces evening sleep increase (ZT9-ZT12) (red line) compared to baseline. (Q–S) Optogenetic activation of ppkAC using CsChrimson (*Ppk-Gal4>Uas-CsChrimson*) causes evening locomotion decrease (ZT9-ZT12) (red bar) compared to baseline. Data (J, M, P, S) were analyzed with Welch's one-way ANOVA (letters a, b, c indicate statistical significance, $P < 0.05$). Data (I, L, O, R) were quantified with paired *t* test. Data (G) were quantified with unpaired *t* test. ns = not significant, *$P < 0.05$, **$P < 0.01$, ***$P < 0.001$, ****$P < 0.0001$. The raw data in this figure including E, F, G, H, I, J, K, L, M, N, O, P, Q, R and S can be found in S1 Data. AC, anterior cell; EM, electron microscopy.

obtained the reconstructed skeletons of the 4 ACs and 6 LPNs and found a close spatial overlap in the dense region of AC axons (Fig 3B and S1 Video), where the AC presynaptic structure and the LPN postsynaptic structure are abundant (Fig 3C).

To assay the role of the AC-LPN circuit in sleep modulation, we first tested 2 different AC drivers to confirm the modulation mechanism from ACs to LPNs. It has been reported that *TrpA1.sh-Gal4* and *Ppk-Gal4* label different ACs (shAC versus ppkAC) [10,16]. We compared the ACs labeled by these 2 drivers using co-staining. The results show that both *TrpA1.sh-Gal4* and *Ppk-LexA* co-label the same ACs (S5A Fig). However, *TrpA1.sh-Gal4* also labels many other non-AC neurons in the brain (S5A Fig). Therefore, we used the Ppk driver in the subsequent study to test the AC-LPN functional connection.

To determine the functional connection from AC to LPN, we performed co-staining and intersection experiments, which revealed that ACs express choline acetyltransferase (ChAT), confirming their identity as cholinergic neurons (S5B and S5C Fig). Furthermore, we observed that optogenetic activation of ACs with CsChrimson by either ppkAC driver (Fig 3D–3G and S2 Video) or shAC driver (S5D–S5F Fig and S3 Video) significantly increased calcium levels in LPNs. This increase could be blocked by the nicotinic acetylcholine receptor (nAChR) antagonist mecamylamine (MEC) (Fig 3E–3G). Together, these data provide evidence that ACs activate LPNs through excitatory acetylcholine signaling.

Both shAC and ppkAC neurons express the high temperature-sensitive TrpA1 channel [10], and shAC has been shown to regulate high-temperature-induced changes in evening peak activity [16]. Since shAC and ppkAC are the same neurons, we hypothesized that manipulation of ACs by the Ppk driver could also influence high-temperature-induced evening peak changes. Indeed, blocking synaptic transmission of ACs using tetanus toxin light chain significantly attenuated the high-temperature-induced increase in evening sleep (Fig 3H–3J) and decrease in evening locomotor peaks (Fig 3K–3M) compared to controls. Furthermore, optogenetic activation of ACs at 23°C was sufficient to mimic the high temperature effect on daytime sleep (Fig 3N–3P) and locomotor profiles of flies (Fig 3Q–3S), suggesting a role for ppkAC in heat-induced changes in daytime activity.

In conclusion, our results provide evidence that high-temperature-sensitive ACs activate LPNs to release AstC to reshape evening activity under high-temperature conditions.

## LPN signal to circadian neuron LNd and inhibit LNd through AstC pathway

The Evening locomotor peak is known to be controlled by E cells, mainly LNd neurons [41]. Our connection analysis revealed that DN1ps, sCPDN3s, and LNds were the major postsynaptic neurons of LPNs. Here, we mainly focused on the neural connection from LPN to LNd as this direct communication may contribute to temperature-dependent Evening peak modulation.

We first analyzed the main postsynaptic neurons of LPNs, including LPNs, DN3A, aME9 [42], and AC (S6A–S6C Fig), which indicates extensive mutual connections between LPNs and major presynaptic neuron types. Then, the data confirmed that all 6 LPNs form synaptic connection with several LNds (Figs 4A and S6D). Meanwhile, the presynaptic neurons of

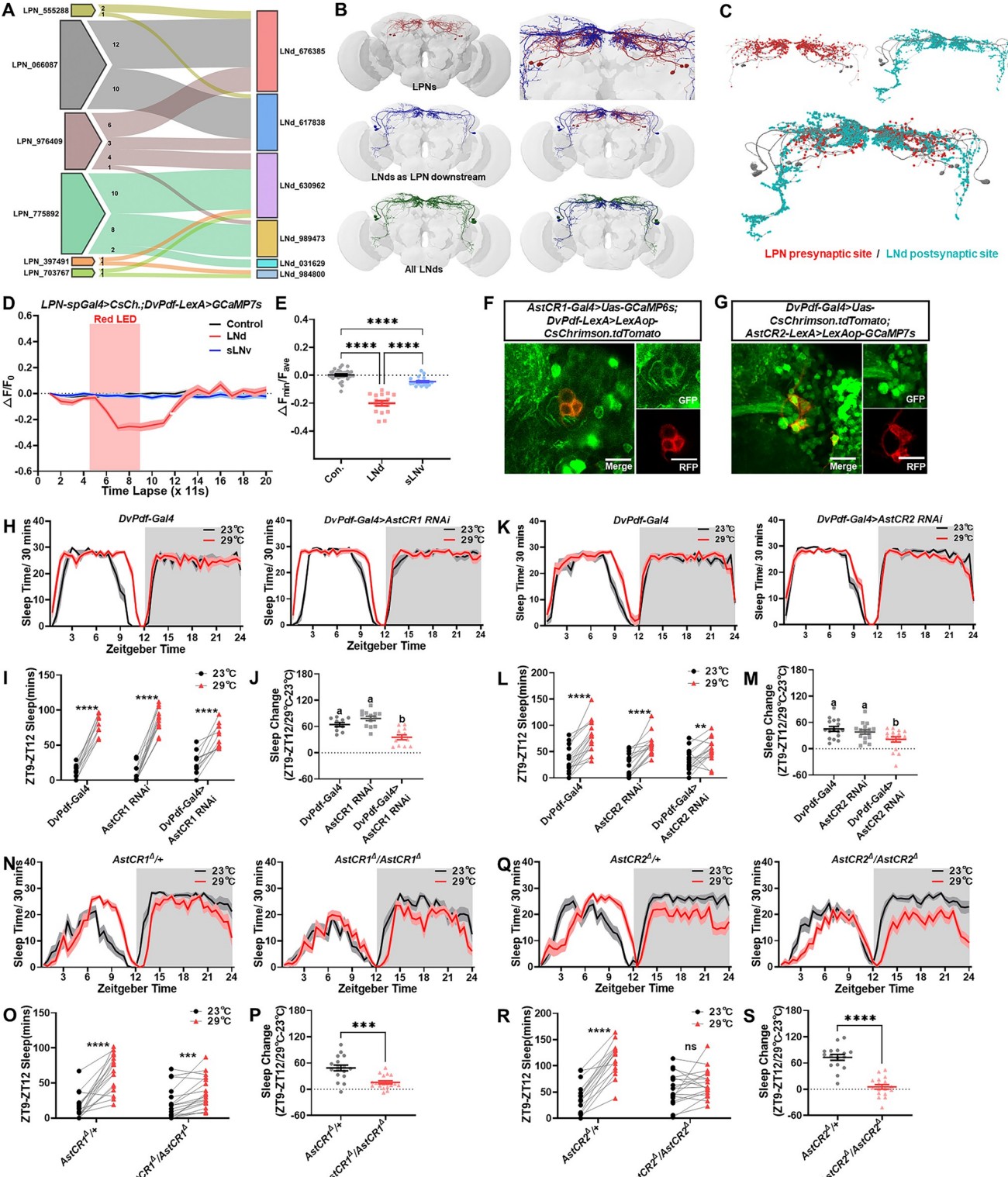

**Fig 4. LPN inhibit LNd through AstC-AstCR pathway to promote evening sleep during temperature increase.** (A) LPN neurons form synaptic connections with some but not all LNds. (B) EM reconstruction showing LPNs (red), LPN-postsynaptic LNd (blue), and all LNds (green), with magnified potential dense synaptic sites (right). (C) Predicted presynaptic sites of LPNs and postsynaptic sites of LPN-postsynaptic LNds in Flywire data set [23]. (D, E) Optogenetic activation of LPNs with CsChrimson (*LPN-spGal4>Uas-CsChrimson*) increases LNd activity (red line) but not sLNv activity (blue line) in vivo. The summarized line graphs (D) and normalized minimum calcium fluorescence (E) are shown. Z-stacks of 10 frames with 1.1 s intervals. (F) AstCR1

neurons (*AstCR1-Gal4>Uas-GCaMP6s*; green) are colocalized with LNds (*DvPdf-LexA>LexAop-CsChrimson.tdTomato*; red). Scale bars, 10 mm. (G) AstCR2 neurons (*AstCR2-LexA>LexAop-GCaMP7s*; green) are colocalized with LNds (*DvPdf-Gal4>Uas-CsChrimson.tdTomato*; red). Scale bars, 10 mm. (H–J) *AstCR1* knockdown in LNds (*DvPdf-Gal4>Uas-AstCR1 RNAi*) blocks the high-temperature-induced evening sleep increase (ZT9-ZT12) (red line, 29°C) compared to baseline (black line, 23°C). Quantification of evening sleep (I) and sleep change (J) shown. (K–M) *AstCR2* knockdown in LNds (*DvPdf-Gal4>Uas-AstCR2 RNAi*) blocks the evening sleep increase (ZT9-ZT12) (red bar, 29°C) compared to baseline (black bar, 23°C). (N–P) The high-temperature-induced evening sleep increase (ZT9-ZT12) (red line, 29°C) is blocked in *AstCR1* mutant flies compared to baseline (black line, 23°C). (Q–S) The high-temperature-induced evening sleep increase (ZT9-ZT12) (red bar, 29°C) is blocked in *AstCR2* mutant flies compared to baseline (black bar, 23°C). Data (J, M) were analyzed using Welch's one-way ANOVA (letters a, b denote statistical significance, $P < 0.05$). Data (I, L, O, R) were quantified with paired $t$ test. Data (E, P, S) were quantified with unpaired $t$ test. ns = not significant, $*P < 0.05$, $**P < 0.01$, $***P < 0.001$, $****P < 0.0001$. The raw data in this figure including D, E, H, I, J, K, L, M, N, O, P, Q, R, and S can be found in S1 Data.

LNds includes LPNs with almost 0.88% (S6E Fig). We also confirmed circadian LNd neurons as downstream neurons of LPNs by *trans*-Tango and PER antibody co-staining (S7A and S7B Fig). The 3D skeleton reconstructions proved those LNds as Cry$^+$ LNds (Fig 4B). LPNs and LNds form a tight spatial overlap in the vicinity of the pars intercerebralis region (Fig 4B and S4 Video), which is also a dense area where LPN output signals and LNds receives signals (Fig 4C) [43,44]. The GRASP signal (GFP Reconstitution Across Synaptic Partners) [45] also indicates a spatial overlap between the fibers of the LPNs and LNds but not the LNvs (S7C Fig). Taken together, these results suggest the synaptic input from LPNs to LNds.

We next investigated the functional connection of the LPN-LNd circuit because of its inhibitory nature based on behavioral data. To this end, we optogenetically activated the LPN and recorded the LNd calcium response. Specifically, LPN activation significantly decreased calcium levels in LNd fibers while has little effect on calcium levels of sLNv fibers (Fig 4D and 4E and S5 Video), consistent with our hypothesis that the LPN-LNd circuit is inhibitory.

Previously, we demonstrated that LPN induces sleep via AstC (Fig 1M–1R). Several papers also reported the role of *AstCR2* in LNds [46,47]. Our next step was to determine whether LPN inhibits LNd through the AstC-AstCR pathway. Co-staining experiments revealed that LNds expresses the relevant receptors for AstC, including *AstCR1* and *AstCR2* (Fig 4F and 4G), but not AstA receptor (S7D Fig). Thus, we proposed that LPN inhibits LNd through the AstC-AstCR pathways, thereby contributing to the regulation of LNd activity at higher temperatures. Consistently, we genetically knocked down *AstCR1* or *AstCR2* in LNds to release LNds from LPN inhibition at high temperature. This manipulation resulted in a significantly attenuated evening sleep increase caused by temperatures elevation compared to controls (Fig 4H–4M). *AstCR1* and *AstCR2* mutations also blocked the evening sleep increase (Fig 4N–4S). Notably, knockdown of *AstCR2* (S7H–S7J Fig) in LNvs with *Pdf-Gal4* reproduce this effect, while knockdown of *AstCR1* (S7E–S7G Fig) did not. This suggests that LPN inhibits LNd through the AstC-AstCR1 pathway, while affecting LNv through AstC-AstCR2 pathway. In other words, interruption of AstC-AstCR pathway reduces the integration of hot information in the circadian circuit.

In conclusion, our results provide evidence that high-temperature-activated ppkAC-LPN circuits inhibit LNd activity, resulting in changes in evening locomotion peak via the AstC-AstCR pathway. This hierarchical regulation ensures that flies can appropriately transmit environmental temperature information to the circadian center and adjust the timing of their major daily activity peak.

## LNd signal back to LPN to form negative feedback circuit

If high temperature activates sleep-promoting LPNs via ppkAC neurons, an increase in overall sleep duration throughout the high-temperature day would be expected. However, our observations revealed a specific increase in evening sleep, indicating a potential heightened excitability of LPNs in response to elevated temperatures during the evening hours. Given the

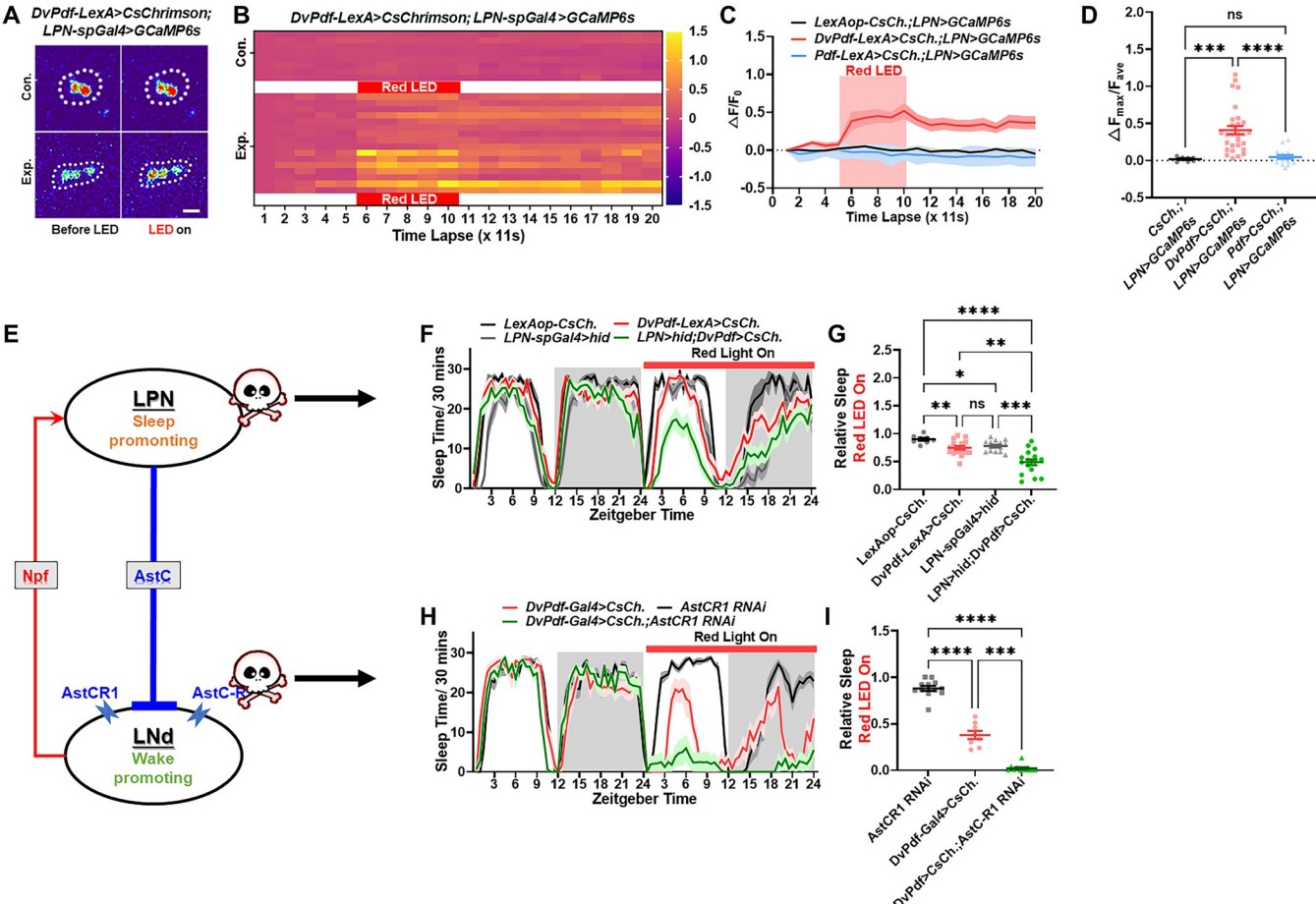

**Fig 5. LNd stimulate LPN to form a circadian feedback circuit.** (A–D) Optogenetic activation of LNds using CsChrimson (*DvPdf-LexA>LexAop-CsChrimson*) increases LPN calcium levels in vivo, with no effect by LNvs activation (*Pdf-LexA>LexAop-CsChrimson*). Heat maps (B), summarized line graphs (C), and normalized maximum calcium fluorescence (D) are shown. Z-stacks of 10 frames with 1.1 s intervals. (E) A working model of LPN-LNd feedback circuit. The red line represents the activation effect and the blue line represents the inhibition effect. If the LPN neurons and AstCR1 receptor in the negative feedback circuit are interfered, it is speculated that the LNd neurons will be affected. (F, G) Optogenetic activation of LNds using CsChrimson (*DvPdf-LexA>LexAop-CsChrimson*) reduces sleep (red line) and this decrease effect was enhanced by LPN inhibition (*DvPdf-LexA>LexAop-CsChrimson*, *LPN-spGal4>Uas-hid*; green line). Quantification of relative second-day sleep compared to baseline (G). (H, I) Optogenetic LNd activation caused sleep decrease (*DvPdf-Gal4>Uas-CsChrimson*, red line) and this decrease effect was enhanced by AstCR1 interference (*DvPdf-Gal4>Uas-CsChrimson*, *Uas-AstCR1 RNAi*; green line). The quantification of relative second day sleep compared with the baseline day (I). Data (D) were quantified with unpaired *t* test. Data (G, I) were analyzed using Welch's one-way ANOVA. ns = not significant, *$P < 0.05$, **$P < 0.01$, ***$P < 0.001$, ****$P < 0.0001$. The raw data in this figure including B, C, D, F, G, H, and I can be found in S1 Data.

documented higher calcium levels in the LNds during the evening [48], we proposed that LNd may activate LPN during this time, with LPN potentially exerting negative feedback on LNds to maintain their activity within an appropriate range.

We firstly analyzed the main postsynaptic neurons of LNds in Flywire database [23], which included LNd, DN3A, DN1p, and sLNv but not LPN (S8A and S8B Fig). To investigate the functional implications of the LNd-LPN connection, we employed optogenetic techniques to activate LNds while concurrently monitoring the neuronal activity of LPNs using GCaMP. Our results demonstrated a significant increase in calcium levels in LPNs following LNd activation with *DvPdf-LexA* while activation of LNvs with *Pdf-LexA* had minimal impact on LPN calcium levels (Fig 5A–5D and S6 Video). Those results clearly demonstrate a negative feedback circuit between LNds and LPNs.

Given that LNds activates LPNs for feedback inhibition, the removal of LPNs' inhibition should render LNds more excitable. This hypothesis was verified by activating LNds while simultaneously obstructing the incoming LPN signal, either through LPN ablation (Fig 5F and 5G) or reducing AstCR1 expression (Fig 5H and 5I) in LNds. Despite variations in the wake-promoting effects between *DvPdf-Gal4>Uas-CsChrimson* and *DvPdf-LexA>LexAop-CsChrimson*, our epistasis results revealed an enhanced ability of LNds to promote wakefulness when LPN input was disrupted. Those data underscore the role of the LNd-LPN negative feedback in maintaining appropriate levels of LNd activation, as the activation of LNds by CsChrimson should override the inhibitory effect of LPNs if there is no excitatory input from LNds to LPNs.

In summary, our findings substantiate the presence of a LNd-LPN negative feedback circuitry. LNd exhibit the capability to activate LPN, while LPN, via the AstC-AstCR pathway, exert a negative modulatory influence on LNd activity. This feedback mechanism plays a crucial role in regulating evening activity under seasonal change in nature.

## LNd-LPN feedback circuit modulate low temperature induced evening activity change

The small number of synapses from LNd to LPN implies that the communication from LNd to LPN is more likely to involve neuropeptides (S8C Fig). This prompted us to investigate the specific neuropeptide responsible for the excitatory LNd-LPN input. It is known that LNds express various neuropeptides, including ITP, sNPF, and NPF [12,38,49–51]. Notably, LPNs express NPF receptors (NPFR) (Fig 6B). Through co-staining and intersection technology, we confirmed the expression of NPF in wake-promoting LNds but not in LNvs (Figs 6A and S9A–S9C). Based on this, we hypothesized that LNds activates LPNs via NPF. Consequently, we reasoned that knockdown of *NPF* in LNds would reduce LPN activation, thus releasing LNds from inhibition, akin to the effects observed with *AstCR1* knockdown in LNd or LPN inhibition. As anticipated, the reduction of NPF in LNds significantly enhanced the wake-promoting effect of LNds (Fig 6C and 6D). Notably, this data also suggests that LNds does not employ NPF to promote activity. Collectively, our data supports the notion that LNds releases NPF to activate LPNs, which in turn negatively feedbacks to LNds via AstC signaling. However, knockdown of *NPF* in LNds and knockdown of *NPFR* in LPNs does not affect the decrease in evening locomotion caused by high temperatures (S9D–S9I Fig), which is reasonable because high temperatures inhibit LNd activity.

To assess the in vivo function of the LNd-LPN negative feedback circuit, we opted to employ cold temperature in LD cycles. This choice was informed by the knowledge that high temperatures suppress LNd activity, while cooling stimulates LNds, inducing a larger evening peak [17]. During this cooling process, the LNd-LPN negative feedback circuit should play a role in constraining LNd activity to an appropriate range. We quantified the total sleep time from ZT6 to ZT12, considering that cold temperatures lead to a substantial decrease in afternoon sleep. As anticipated, moderate low temperatures induced a visible evening peak in control flies. However, the knockdown of *NPF* in LNds intensified the decrease in evening sleep (Fig 6E–6G) and amplified the increase in evening peaks (Fig 6H–6J) at lower temperatures. Similarly, the knockdown of *NPFR* in LPN exhibited a similar behavioral effect (Fig 6K–6P) at moderately lower temperatures.

In conclusion, our findings emphasize the significance of the circadian circuit between LPN and LNd in regulating evening locomotion under varying temperature conditions. This circuit ensures the maintenance of an appropriate level of evening locomotion, which is crucial for the energy balance and overall survival.

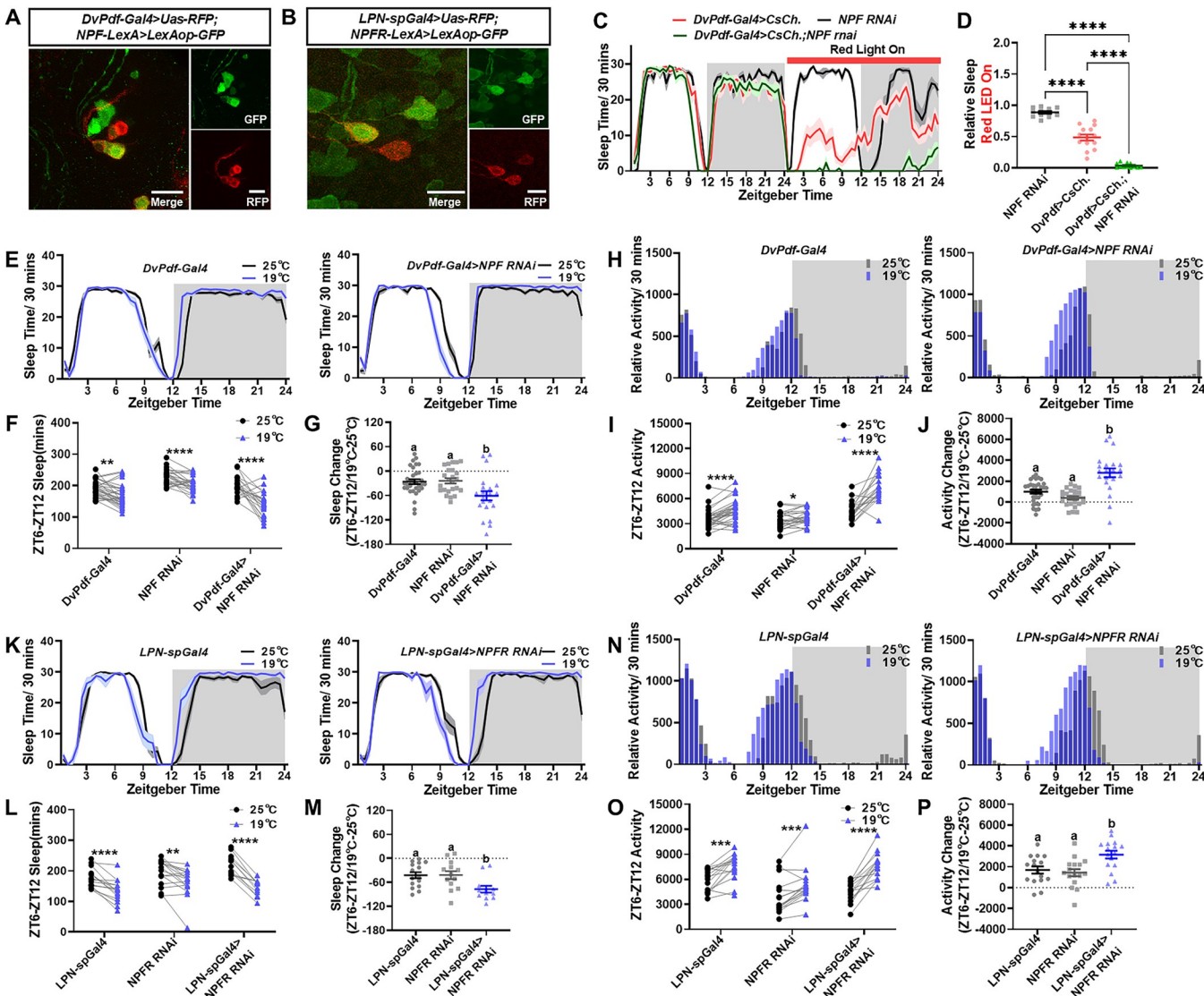

**Fig 6. NPF-NPFR pathway in LNd-LPN feedback circuit fine-tune cooling induced locomotion increase.** (A) NPF neurons (*NPF-LexA>LexAop-GFP*; green) are colocalized with LNds (*DvPdf-Gal4>Uas-RFP*; red). Scale bar, 10 µm. (B) NPFR neurons (*NPFR-LexA>LexAop-GFP*; green) are colocalized with LPNs (*LPN-spGal4>Uas-RFP*; red). Scale bar, 10 µm. (C, D) Optogenetic activation of LNds using CsChrimson (*DvPdf-Gal4>Uas-CsChrimson*, red line) decreases sleep and this decrease effect was enhanced by NPF RNAi (*DvPdf-Gal4>Uas-CsChrimson; Uas-NPF RNAi*; green line). Relative second-day sleep compared to baseline quantified (D). (E–G) *NPF* knockdown in LNds (*DvPdf-Gal4>Uas-NPF RNAi*) blocks low temperature (19˚C) induced decrease in evening sleep (ZT6-ZT12). Evening sleep (F) and sleep change (G) were quantified. (H–J) *NPF* knockdown in LNds (*DvPdf-Gal4>Uas-NPF RNAi*) blocks low temperature (19˚C) induced increase in evening locomotion. Relative evening locomotion (I) and locomotion change (J) were quantified. (K–M) *NPFR* knockdown in LPNs (*LPN-spGal4>Uas-NPFR RNAi*) blocks low temperature (19˚C) induced decrease in evening sleep (ZT6-ZT12). Evening sleep (L) and sleep change (M) were quantified. (N–P) *NPFR* knockdown in LPNs (*LPN-spGal4>Uas-NPFR RNAi*) blocks low temperature (19˚C) induced increase in evening locomotion. Relative evening locomotion (O) and locomotion change (P) were quantified. Data (G, J, M, P) were analyzed with Welch's one-way ANOVA (letters a, b denote significance, *P* < 0.05). Data (F, I, L, O) were quantified with paired *t* test. Data (D) were analyzed with unpaired *t* test. *\*P* < 0.05, *\*\*P* < 0.01, *\*\*\*P* < 0.001, *\*\*\*\*P* < 0.0001. The raw data in this figure including C, D, E, F, G, H, I, J, K, L, M, N, O, and P can be found in S1 Data.

## Discussion

In summary, our study has elucidated a circadian circuit that dynamically responds to temperature changes, thereby modulating evening activity in *Drosophila*. This discovery highlights the pivotal role of the LNd-LPN feedback circuit in enhancing the adaptability of wake-sleep patterns across different seasons. In nature, diurnal animals such as fruit flies, fishes, birds,

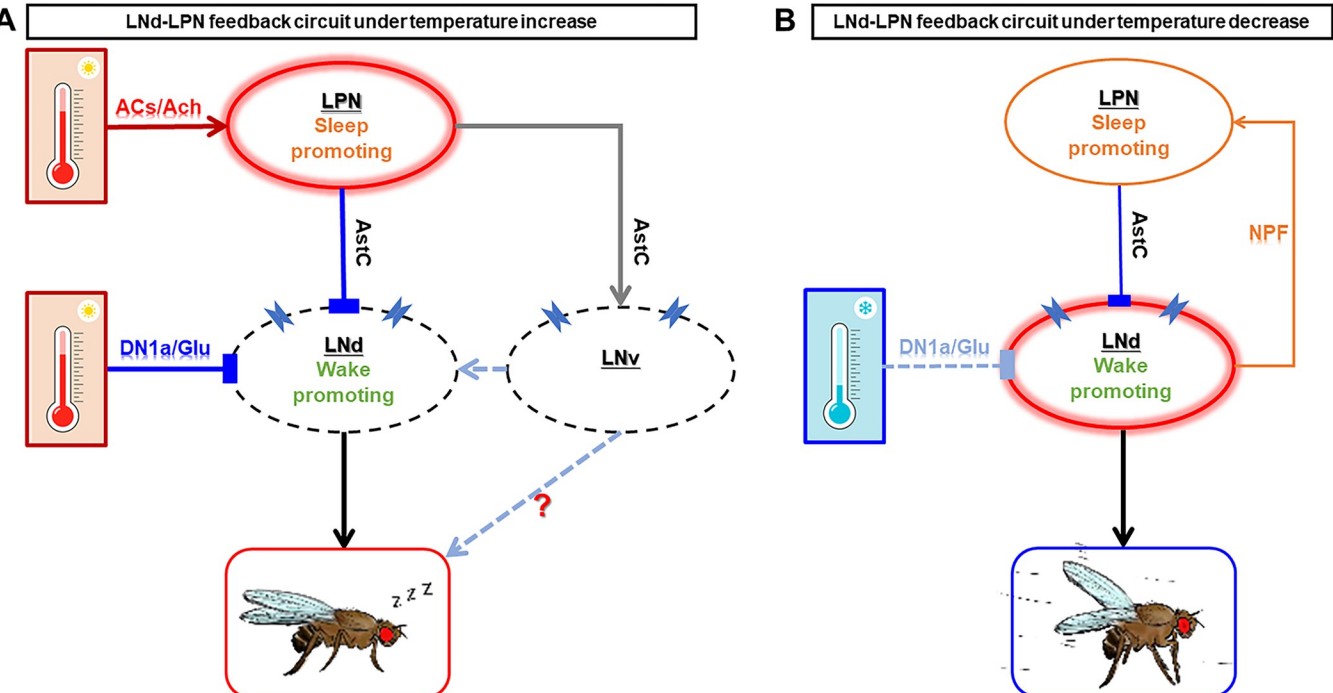

**Fig 7. A model for LPN-mediated regulation of sleep behavior.** (A) A model for AC-LPN-LNd circuit to induce evening sleep increase under high temperature. When temperature is changed, organism regulates locomotion-sleep balance for better survival. LPNs are more sensitive to temperature increase due to the input from high temperature sensitive AC cells. DN1as and LPNs will be activated by temperature elevation and LPNs then release AstC to inhibit LNds, which dominate evening sleep/locomotion. (B) A model for LNd-LPN feedback circuit to limit overactivity induced by over-activation of LNds under low temperature. LNds are activated by low temperature sensitive DN1a cells, then promote evening locomotion in cold conditions. At the same time, LNds signal to LPNs via NPF-NPFR pathway to prevent excessive movement, which is also dangerous under low-temperature environment. The model was modified and integrated based on our previous publication [17]. AC, anterior cell.

and mammals demonstrate an ability to adjust their movement and energy expenditure in response to environmental changes [17,52–54]. Central to this adaptive strategy is the maintenance of activity at an optimal level, which is crucial for efficient energy management and, ultimately, increased survival prospects.

The electron microscopy reconstruction of the fruit fly brain structure has greatly facilitated research on neural circuits. Based on this, we analyzed the LPN as a circadian neuron that receives input from environmental factors such as temperature and coordinates inputs from other circadian neurons to dynamically regulate sleep behavior under varying environmental conditions. Under natural conditions, particularly at dusk, the activity of LNd gradually increases, promoting alertness and locomotion for activities such as feeding [7,55–57]. The sensitivity of LNd to temperature is instrumental in efficient energy management and survival. The high-temperature-induced activation of LPN by ACs and activation of DN1a by TPNs predominantly inhibit LNd, resulting in inhibition of evening locomotion and better energy preservation to elevated temperatures (Fig 7A). Conversely, cooling-induced activation of LNd predominantly releases and concurrently activates LPN to provide feedback inhibition, thereby restricting locomotion within an appropriate range and facilitating adaptation to lower temperatures (Fig 7B).

Temperature-sensory neurons in *Drosophila* have been identified based on the expression of temperature-sensitive proteins such as TrpA1, IR25a, and Brvs [18,19,58]. Previous studies have shown that shAC and ppkAC neurons express TrpA1 channels but belong to distinct cellular populations; this distinction is evident as *ppk-Gal80* can inhibit ppkAC expression while

leaving shAC expression unaffected [10]. In our investigation, we assessed the co-localization of shAC and ppkAC neurons and found that they are indeed the same neurons. Additionally, both shAC and ppkAC neurons can activate LPN. This discrepancy may arise from the differential inhibitory strength of *ppk-Gal80*, which is sufficient to inhibit *ppk-Gal4* but not *TrpA1. sh-Gal4*.

Temperature fluctuations modulate the activity of circadian neurons [11,17]. Recent studies have uncovered direct synaptic connections between temperature-sensing neurons and circadian neurons. Specifically, DN1a neurons receive cold signals from TPN-II neurons, while DN1p and LPN neurons receive thermal signals from ACs and TPN-IV neurons [14–16]. However, these studies have not thoroughly explored how these 2 systems coordinate to control sleep behavior. The circadian neuron network is intricate and highly interconnected [43,44]. Different sensory inputs appear to target specific circadian neurons based on their molecular and connectomic properties [38]. Temperature signals primarily input into a limited number of circadian neurons, such as DN1a (2 cells) and LPN (3 cells), depending on whether the cues are cold or thermal [14–17]. Although DN1a neurons respond to both high and low temperatures, they use glutamate as a neurotransmitter to regulate the evening peak at low temperatures [17]. Conversely, LPN neurons fine-tune the evening peak using neuropeptides at both high and low temperatures, but their effect at low temperatures relies on the DN1a-LNd circuit.

Thus, the regulation of the evening activity peak varies with temperature, with DN1a primarily controlling this peak at low temperatures, and LPN primarily controlling it at high temperatures. DN1a utilizes neurotransmitters, whereas LPN relies on neuropeptides for temperature regulation. We hypothesize that this difference in mechanisms may lead to varying time scales in sleep-wake regulation by temperature. This differential use of neurotransmitters and neuropeptides may also explain why EM connectomic data indicate abundant synaptic connections from DN1a to LNd, but fewer synaptic connections between LPN and LNd. The neuropeptide-dependent communication in the LPN-LNd pathway largely operates independently of synaptic vesicle release. Furthermore, LPNs express the excitatory neurotransmitter acetylcholine. The inhibitory effect of AstC can be maintained despite the presence of excitatory neurotransmission. Another notable finding from our study is that the circuit regulating evening locomotion patterns, regardless of temperature changes, converges on LNd, highlighting its pivotal role as the principal circadian neuron controlling evening peaks [41].

## Material and methods

### Fly stocks and experiments

All flies were reared on standard cornmeal/agar medium supplemented with yeast at 25°C in 12:12 light-dark (LD) cycles. Adult male flies aged 3 to 10 days were used for behavior and imaging experiments. For optogenetic experiments, adult flies were collected and kept in weak light condition on standard food supplemented with 800 mM all-trans-retinal. Flies for Multi-Color FlpOut and TrpA1/Shib[TS] experiments were raised at 18 to 20°C. The S1 Table provides a list of fly lines used in this study.

As previously described [59], all-trans-retinal (ATR) powder (Sigma) was dissolved in 100% alcohol to prepare a 100 mM stock solution. For ATR food used in the optogenetic experiments, 240 μl of stock solution was diluted in 30 ml of 5% sucrose and 1% agar medium to prepare final food with approximate 800 μm ATR. Flies aged 3 to 5 days were transferred to ATR food for at least 3 days before optogenetic experiments.

## Sleep recording and quantification

The Flybox system for the optogenetics and video recording is described in detail [60]. Adult male flies (aged 3 to 7 days) were placed in white 96-well microfluor 2 plates (Fishier) with 300 μl of food (5% sucrose and 1% agar with or without 800 mM ATR). The flyboxes were kept in an incubator to maintain precise temperature control. A webcam tracked fly locomotion at 10-s intervals, and the data were analyzed using the Sleep and Circadian Analysis MATLAB Program (48) for sleep analysis [61]. A sleep episode was defined as inactivity lasting at least 5 min [62,63]. All experiments were replicated at least 3 times. All data was plotted in GraphPad Prism 9.

For optogenetic experiments, flies were entrained on the ATR food under 12-h LD cycles for 3 days. At ZT0 on the fourth day, light pulses (627 nm for CsChrimson) (approximately 0.1 m/Wmm2) were used to optogenetic manipulation. For experiments involving temperature shifts, flies were entrained to 12-h light-dark (LD) cycles for 3 days at 25°C/23°C, then exposed to 19°C/29°C for 1 day. Temperature changes occurred at ZT0 on the fourth day. Sleep time from ZT6-ZT12 was quantified because cooling induces forward locomotion in the evening. Sleep time from ZT9-ZT12 was quantified because heating induces decreased locomotion in the evening.

## Immunohistochemistry

Adult flies were dissected in PBS and fixed on ice with 4% paraformaldehyde and 0.008% Triton X-100 in PBS for 30–45 min. After 3 washes of 10 min each in PBST (PBS plus 0.5% Triton-X), the samples were incubated with primary antibodies overnight at 4°C. The brains were then washed 3 times for 10 min each with PBST and incubated in secondary antibodies for 2 h at room temperature. After washing with PBST, the samples were mounted in Vectashield Mounting Medium (Vector Laboratories) and viewed under a laser scanning confocal microscope (Olympus) at 10×, 20×, or 60× magnification using 2 or 2.5 μm sections. For PER staining, flies were dissected at ZT 21 after 3 to 5 days of LD entrainment.

The antibodies used in this study were listed in S1 Table.

## Sparse labeling with MultiColor FlpOut

Male adult brains from LPN-spGal4>nSyb-Flp; Uas-frt-stop-frt-myr::smGdP-HA,Uas-frt-stop-frt-myr::smGdP-V5,Uas-frt-stop-frt-myr::smGdP-FLAG flies were dissected for immunostaining. Brains were labeled to visualize 3 markers (HA, V5, and FLAG) as described [32]. The immunostaining steps have been described above. Approximately 5% of single neurons are marked by one of the 3 tags.

## Living calcium imaging with two-photon microscope

All imaging data was acquired in vivo. Flies were anesthetized with CO2 and their heads were fixed on a customized chamber with UV glue. A small window was opened on the anterior side of the cuticle using sharp forceps under extracellular saline solution ECS (103 mM NaCl, 3 mM KCl, 5 mM N-tris(hydroxymethyl) methyl-2-aminoethane-sulfonic acid, 10 mM trehalose, 10 mM glucose, 2 mM sucrose, 26 mM NaHCO3, 1 mM NaH2PO4, 1.5 mM CaCl2, and 4 mM MgCl2, adjusted to 275 mOsm, pH equilibrated near 7.1 to 7.3). The ECS containing the corresponding concentrations of MK801 (200 μm) and MEC (10 μm) was perfused for at least 10 min before the optogenetic manipulation [39]. For Muscle 16 and obstructing trachea were removed to prevent brain movement during imaging. For optogenetic activation, a 630

nm LED illuminated the cuticle window with a light intensity of 1 to 2 mW. Images were typically acquired at a resolution of 512 × 512 pixels and a frame rate of approximately 0.9 Hz.

ROIs were analyzed using Fiji [64]. The fluorescence change was calculated as: $\triangle F/F0 = (Ft-F0)/F0$, $\triangle Fmax/Fave = (Fmax–Fave)/Fave$, and $\triangle Fmin/Fave = (Fmin–Fave)/Fave$, where Ft is the fluorescence at time point t. F0 is the fluorescence at time 0, Fmax is the maximum fluorescence during optogenetic manipulation, Fmin is the minimum fluorescence during optogenetic manipulation, and Fave is the average fluorescence before optogenetic manipulation.

### Flywire connectome analysis and visualization

Data were visualized using Graphpad v9.0.0 or OriginPro 2024. All data for connectome analysis is downloaded with FAFBseg tools (https://pypi.org/project/fafbseg/). For the presynaptic or postsynaptic analysis for neuron types, only neurons with ≥5 synapse is included. Reconstructions and visualization are performed with FAFBseg or neuVid (https://github.com/connectome-neuprint/neuVid). The IDs of related circadian neurons are acquired from recent bioRxiv paper [29] and IDs of other related neurons are proved as S2 Table [23].

### Quantification and statistical analysis

The data were assessed for normal distribution using the Wilks–Shapiro test. If the data followed a normal distribution, 2-tailed, unpaired Student's *t* tests and paired *t* tests were conducted. For quantification more than 3 groups, one-way analysis of variance (ANOVA) with a Tukey–Kramer HSD Test were adopted. The data is presented as a scatter dot plot with mean bars, and the error bars in sleep quantifications represent the standard error of the mean (SEM) while the error bars in imaging experiments represent the standard deviation (SD). Differences between groups were considered significant if the *P*-value was less than 0.05 ($P < 0.05$).

### Supporting information

**S1 Data. Raw data underlying quantification figures mentioned.**
(XLSX)

**S1 Table. The genotypes and sources of all fly lines and materials in this study.**
(XLSX)

**S2 Table. All Flywire IDs of related neurons in this study.**
(XLSX)

**S1 Fig. (Related to Fig 1): LPN promote evening sleep via AstC under high temperature condition.** (A, B) LPN activation via the thermogenetic dTrpA1 channel (*LPN-spGal4>Uas-dTrpA1*; red line) significantly increases sleep compared to baseline. (C, D) LPN activation via the thermogenetic dTrpA1 channel (*LPN-spGal4>Uas-dTrpA1*) significantly decreases daily locomotion compared to baseline (black bar). (E) AstA neurons (*AstA-LexA>LexAop-GFP*; green) are colocalized with LPNs (*LPN-spGal4>Uas-RFP*; red). Scale bar, 10 μm. (F) ChAT neurons (*ChAT-LexA>LexAop-GFP*; green) are colocalized with LPNs (*LPN-spGal4>Uas-RFP*; red). Scale bar, 10 μm. (G, H) Intersection neurons between *VGLUT-LexA* (G) or *ChAT-LexA* (H) and *LPN-spGal4* are labeled with GFP. Intersection with *ChAT-LexA* clearly marks LPNs (arrows). Scale bar, 100 μm. (I, J) Activation of intersection-labeled LPNs (red line) has a sleep-promoting effect similar to *LPN-spGal4* activation (yellow line). All data were analyzed using Welch's one-way ANOVA with multiple comparisons, and letters a, b, and c

indicate significant differences, $P < 0.05$. ns = not significant, ***$P < 0.001$, ****$P < 0.0001$. The raw data in this figure including A, B, C, D, I, and J can be found in S1 Data.
(TIF)

**S2 Fig. (Related to Fig 2): Connectomics analysis of LPN neurons with circadian neurons.** (A) EM reconstruction of LPN subtypes CB1215 and CB2857 in Flywire database. (B) EM reconstruction of recently annotated circadian cell types in Flywire database. (C, D) Quantification of presynaptic inputs to LPNs from all annotated circadian neurons (C) and postsynaptic outputs from LPNs to all annotated circadian neurons (D). Circle size represents the number of synapses and each circle means one single neuron.
(TIF)

**S3 Fig. (Related to Fig 2): The circadian DN1p and sCPDN3 signal to LPN.** (A, B) *VGLUT* knockdown in DN1ps (*R18H11-Gal4>Uas-VGLUT RNAi*) reduces daily sleep duration. Sleep patterns (A) and total sleep time (B) are shown. (C) NMDAR1 neurons (*NMDAR1-Lex-A>LexAop-GFP*; green) are colocalized with LPNs (*LPN-spGal4>Uas-RFP*; red). Scale bar, 10 μm. (D, E) Optogenetic activation of DN1ps (*R18H11-LexA>LexAop-CsChrimson*) increases LPN calcium levels in vivo, and Glu antagonist-MK801 blocks this activation. Heat maps (D) and normalized maximum calcium levels (E) are shown. Z-stacks of 10 frames with 1.1 s intervals. (F) Confocal images of sCPDN3-spGal4 (*sCPDN3-spGal4>Uas-GFP*) showing brain and ventral nerve cord. Scale bar, 100 μm. (G, H) Optogenetic activation of sCPDN3 (*sCPDN3-spGal4>Uas-CsChrimson*; red line) significantly increases sleep compared to baseline. (I, J) Optogenetic activation of sCPDN3 (*sCPDN3-spGal4>Uas-CsChrimson*) significantly decreases daily locomotion compared to baseline (black bar). Data (E) were analyzed with an unpaired *t* test. Data (B, H, J) were analyzed using Welch's one-way ANOVA with multiple comparisons, and letters a and b denote significant differences, $P < 0.05$. The raw data in this figure including A, B, D, E, G, H, I, and J can be found in S1 Data.
(TIF)

**S4 Fig. (Related to Fig 3): The main downstream neurons of ACs in Flywire database.** (A) Top 50 postsynaptic neurons of ACs from the FlyWire Data set (v783). (B) EM reconstruction of AC postsynaptic neurons from panel A.
(TIF)

**S5 Fig. (Related to Fig 3): The TrpA1.shAC colocolized with ppkAC and its activation promote calcium levels of LPNs.** (A) Ppk neurons (*Ppk-LexA>LexAop-CsChrimson.tdTomato*; red) are colocalized with TrpA1.sh neurons (*TrpA1.sh-Gal4>Uas-GCaMP6s*; green). Scale bar, 50 μm. (B, C) ACs are cholinergic. ChAT neurons (*ChAT-LexA>LexAop- GCaMP7s*; green) are colocalized with Ppk neurons (*Ppk-Gal4>Uas- CsChrimson.tdTomato*; red) (B). AC neurons are clearly labeled by intersection between *ChAT-LexA* and *Ppk-Gal4* (arrows) (C). Scale bar, 100 μm. (D–F) Optogenetic activation of TrpA1.sh neurons with CsChrimson (*TrpA1.sh-Gal4>Uas-CsChrimson*) promotes LPN calcium levels in vivo. Heat maps (D), a summarized line graph (E) and normalized calcium levels (F) are shown. Z-stacks of 10 frames with 1.1 s intervals. Unpaired *t* test. **$P < 0.01$. The raw data in this figure including D, E, and F can be found in S1 Data.
(TIF)

**S6 Fig. (Related to Fig 4): The main downstream neurons of LPN in Flywire database.** (A) Top 50 postsynaptic neurons of LPNs based on FlyWire Data set (v783). (B) Percentage of output from LPNs to various neuron types with ≥5 synapses, as predicted using Flywire data set. (C) EM reconstruction of LPN postsynaptic neurons from panel B. (D) Synaptic connections

from each LPN to each LNd. (E) Percentage of input to LNds from different neuron types with ≥5 synapses, as predicted using Flywire data set.
(TIF)

**S7 Fig. (Related to Fig 4): LPN signals to LNd and LNv with AstC-AstCR pathway.** (A) LPNs (red) and their postsynaptic targets (green) shown with *LPN-spGal4>Uas-RFP; trans-Tango,QUas-GFP*. Arrows indicate LNd-like cells. Scale bar, 50 µm. (B) Co-staining of PER (red) with all LPN downstream targets (*LPN-spGal4>transTango,Quas-GFP*; green), highlighting the circadian neuron identity. Scale bar, 10 µm. (C) LPNs form synaptic spatial overlaps with LNds (*LPN-spGal4>Uas-sp1-10;DvPdf-LexA>LexAop-GFP11*; left) but not with sLNv (*LPN-spGal4>Uas-sp1-10;Pdf-LexA>LexAop-GFP11*; right). Scale bar, 50 µm. (D) AstAR1 neurons (*AstAR1-LexA>LexAop-GCaMP7s*; green) did not colocalized with LNds (*DvPdf-Gal4>Uas-CsChrimson.tdTomato*; red). Scale bar, 10 µm. (E–G) Evening sleep increase (ZT9-ZT12) at high temperature (29˚C, red line) compared to baseline (23˚C, black line) was unaffected by *AstCR1* knockdown in LNvs (*Pdf-Gal4>Uas-AstCR1 RNAi*). Quantification of evening sleep (F) and sleep change (G) before and after temperature increase. (H–J) Evening sleep increase (ZT9-ZT12) at high temperature (29˚C, red line) compared to baseline (23˚C, black line) was blocked by *AstCR2* knockdown in LNvs (*Pdf-Gal4>Uas-AstCR2 RNAi*). Quantification of evening sleep (I) and sleep change (J) before and after temperature increase. Data (G, J) were analyzed using Welch's one-way ANOVA with multiple comparisons; letters a, b, and c denote significant differences, $P < 0.05$. Data (F, I) were analyzed with paired $t$ test. \*\*\*$P < 0.001$, \*\*\*\*$P < 0.0001$. The raw data in this figure including E, F, G, H, I, and J can be found in S1 Data.
(TIF)

**S8 Fig. (Related to Fig 5): The main downstream neurons of LNd in Flywire database.** (A) Top 50 postsynaptic neurons of LNds based on FlyWire Data set (v783). (B) Percentage of output from LNds to various neuron types with ≥5 synapses, as predicted using Flywire data set. (C) Synaptic connections from each LNd to each LPN.
(TIF)

**S9 Fig. (Related to Fig 6): NPF in LNd and NPFR in LPN are not required to high-temperature-induced evening sleep decrease.** (A) Intersection neurons between *DvPdf-LexA* and *NPF-Gal4* labeled LNds (arrows). Scale bar, 100 µm. (B, C) Optogenetic activation with CsChrimson of the intersectional neurons between *DvPdf-LexA* and *NPF-Gal4* (red lines) causes a strong sleep-inhibiting effect. (D–F) High temperature (29˚C) induced evening sleep increase (ZT9-ZT12) was not blocked by *NPF* knockdown in LNds (*DvPdf-Gal4>Uas-NPF RNAi*). Quantification of evening sleep (E) and sleep change (F) before and after temperature increase. (G–I) High temperature (29˚C) induced evening sleep increase (ZT9-ZT12) was not blocked by *NPFR* knockdown in LPNs (*LPN-spGal4>Uas-NPFR RNAi*). Quantification of evening sleep (H) and sleep change (I) before and after temperature increase. Data (C, F, I) were analyzed using Welch's one-way ANOVA, and same letter a denote no significant difference. Data (E, H) were analyzed with paired $t$ test. \*$P < 0.05$, \*\*$P < 0.01$, \*\*\*$P < 0.001$, \*\*\*\*$P < 0.0001$. The raw data in this figure including B, C, D, E, F, G, H, and I can be found in S1 Data.
(TIF)

**S1 Video. (Related to Fig 3): Animation of ACs and LPNs in Flywire database.**
(AVI)

**S2 Video. (Related to Fig 3): Optogenetic activation of ppkAC promotes the calcium levels of LPNs.** Video showing optogenetic stimulation of ppkAC and its effect on calcium levels in LPN cell bodies. Fly genotype: *LPN-spGal4>Uas-GCaMP6s; LexAop-CsChrimson.tdTomato* and *LPN-spGal4>Uas-GCaMP6s; Ppk-LexA>LexAop-CsChrimson.tdTomato*.
(AVI)

**S3 Video. (Related to Fig 3): Optogenetic activation of TrpA1.shAC promotes the calcium levels of LPNs.** Video showing optogenetic stimulation of shAC and its effect on calcium levels in LPN cell bodies. Fly genotype: *Uas-CsChrimson.tdTomato; R65D05-LexA>LexAop-GCaMP6s* and *TrpA1.sh-Gal4>Uas-CsChrimson.tdTomato; R65D05-LexA>LexAop-GCaMP6s*.
(AVI)

**S4 Video. (Related to Fig 4): Animation of LPNs and LNds in Flywire database.**
(AVI)

**S5 Video. (Related to Fig 4): Optogenetic activation of LPNs inhibits the calcium levels of LNds but not sLNvs.** Video showing optogenetic stimulation of LPNs and its effect on calcium levels in LNd terminals (dotted line) but not in sLNv terminals. Fly genotype: *Uas-CsChrimson.tdTomato; DvLexA>LexAop-GCaMP7s* and *LPN-spGal4>Uas-CsChrimson.tdTomato; DvLexA>LexAop-GCaMP7s*.
(AVI)

**S6 Video. (Related to Fig 5): Optogenetic activation of LNds promotes the calcium levels of LPNs.** Video showing optogenetic stimulation of LNds and its effect on calcium levels in LPN cell bodies (dotted line). Fly genotype: *LPN-spGal4>Uas-GCaMP6s; LexAop-CsChrimson.tdTomato* and *LPN-spGal4>Uas-GCaMP6s; DvLexA>LexAop-CsChrimson.tdTomato*.
(AVI)

## Acknowledgments

We thank Dr. Zhefeng Gong, Dr. Chang Liu, Dr. Yi Rao, Dr. Woo Jae Kim, Dr. Yufeng Pan, Dr. Junhai Han, Dr. Wei Zhang, the Vienna *Drosophila* Resource Center, and the Bloomington *Drosophila* Stock Center for sharing the stocks and reagents. We are grateful to Shuangshuang Liu for the technical support in the Core Facilities, Zhejiang University School of Medicine. We are grateful to Sanhua Fang and Dan Yang for the technical support in the Core Facilities, Zhejiang University School of brain science and brain medicine. We thank Xinling Wang for providing the hand-drawn cartoons featured in this manuscript.

## Author Contributions

**Conceptualization:** Xin Yuan, Hailiang Li, Fang Guo.

**Data curation:** Xin Yuan, Fang Guo.

**Formal analysis:** Fang Guo.

**Funding acquisition:** Fang Guo.

**Investigation:** Fang Guo.

**Methodology:** Xin Yuan, Hailiang Li, Fang Guo.

**Project administration:** Fang Guo.

**Resources:** Fang Guo.

**Software:** Xin Yuan, Fang Guo.

**Supervision:** Xin Yuan, Fang Guo.

**Validation:** Xin Yuan, Fang Guo.

**Visualization:** Xin Yuan, Fang Guo.

**Writing – original draft:** Xin Yuan, Fang Guo.

**Writing – review & editing:** Xin Yuan, Fang Guo.

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
