## [Editor Report · Decision Letter 0]

2 Oct 2024

Dear Dr fang, 

Thank you for submitting your manuscript entitled "A flexible circadian neuropeptidergic feedback circuit remodels sleep-wake patterns under temperature changes" for consideration as a Research Article by PLOS Biology.

Your manuscript has now been evaluated by the PLOS Biology editorial staff as well as by an academic editor with relevant expertise and I am writing to let you know that we would like to send your submission out for external peer review.

Once your full submission is complete, your paper will undergo a series of checks in preparation for peer review. After your manuscript has passed the checks it will be sent out for review. To provide the metadata for your submission, please Login to Editorial Manager (https://www.editorialmanager.com/pbiology) within two working days, i.e. by Oct 04 2024 11:59PM.

Kind regards,

Suzanne

Suzanne De Bruijn, PhD

Associate Editor

PLOS Biology

sbruijn@plos.org

---

## [Editor Report · Decision Letter 1]

18 Oct 2024

Dear Dr Guo,

Thank you for your patience while we considered your revised manuscript "A flexible circadian neuropeptidergic feedback circuit remodels sleep-wake patterns under temperature changes", based on the reviews from Science Advances, for publication as a Research Article at PLOS Biology. This revised version of your manuscript has been evaluated by the PLOS Biology editors, and the Academic Editor.

Based on our Academic Editor's assessment of your revision, we are likely to accept this manuscript for publication, provided you satisfactorily address the following data and other policy-related requests.

1) Please change your title to: “Temperature cues are integrated in a flexible circadian neuropeptidergic feedback circuit to remodel sleep-wake patterns in flies”

2) Please provide the raw data underlying the figures, and mention in the legends where these data can be found. Please also ensure you include a proper reference to the FlyWire dataset in the legends. You could provide these raw data in a file called S1_data. 

We need data for: 1D, D’, E’ E’ F, F’ G’ G, I, I’, J, J’; 3D’’, D’’’ E, E’ F F’ F’’ G, G’ G’’ H’, H’’ ;4D, D’, G, G’, H, H’ H’’ I, I’ J’ ;5A’’, A’’’, C, C’, D, D’ ;6C, C’, D, D’, D’’ E, E’, F, F’ G, G’, G’’; S1A, A’, B, B’, F, F’ ;S3A, A’, B’ E, E’ F, F’ ;S5C’, C’’ ;S7E, E’, E’’, F, F’, F’’ ;S9B, B’, C, C’, C’’, D, D’, D''

3) Please use continuous alphabetical labels for the figure panels, instead of the apostrophes.

We expect to receive your revised manuscript within two weeks. 

*Published Peer Review History*

*Press*

Sincerely,

Suzanne

Suzanne De Bruijn, PhD, 

Associate Editor

sbruijn@plos.org

PLOS Biology

DATA POLICY:

[Figs….]

CODE POLICY

Per journal policy, if you have generated any custom code during the course of this investigation, please make it available without restrictions. Please ensure that the code is sufficiently well documented and reusable, and that your Data Statement in the Editorial Manager submission system accurately describes where your code can be found. [IF APPLICABLE: As the code that you have generated to XXX is important to support the conclusions of your manuscript, its deposition is required for acceptance.]

---

## [Editor Report · Decision Letter 2]

30 Oct 2024

Dear Dr Guo,

Thank you for the submission of your revised Research Article "Temperature cues are integrated in a flexible circadian neuropeptidergic feedback circuit to remodel sleep-wake patterns in flies" for publication in PLOS Biology. On behalf of my colleagues and the Academic Editor, Paul Shaw, I am pleased to say that we can in principle accept your manuscript for publication, provided you address any remaining formatting and reporting issues. These will be detailed in an email you should receive within 2-3 business days from our colleagues in the journal operations team; no action is required from you until then. Please note that we will not be able to formally accept your manuscript and schedule it for publication until you have completed any requested changes.

IMPORTANT: I've asked my colleagues to include the following request among their own: "We note that the legend of Fig 2 only cites the original Nat Methods FlyWire paper. Please additionally include the FlyWire URL (https://flywire.ai/) and provide the numerical values underlying Figs 2ABDEFG, as an additional sheet in S1 Data".

PRESS

Sincerely, 

Suzanne

Suzanne De Bruijn, PhD 

Associate Editor

PLOS Biology

sbruijn@plos.org